# Chemokines modulate glycan binding and the immunoregulatory activity of galectins

Lucía Sanjurjo [1], Iris A. Schulkens[1], Pauline Touarin[2], Roy Heusschen[1], Ed Aanhane[1], Kitty C. M. Castricum[3], Tanja D. De Gruijl[1], Ulf J. Nilsson [4], Hakon Leffler [5], Arjan W. Griffioen[1], Latifa Elantak[2], Rory R. Koenen[6] & Victor L. J. L. Thijssen [1,3✉]

Galectins are versatile glycan-binding proteins involved in immunomodulation. Evidence suggests that galectins can control the immunoregulatory function of cytokines and chemokines through direct binding. Here, we report on an inverse mechanism in which chemokines control the immunomodulatory functions of galectins. We show the existence of several specific galectin-chemokine binding pairs, including galectin-1/CXCL4. NMR analyses show that CXCL4 binding induces changes in the galectin-1 carbohydrate binding site. Consequently, CXCL4 alters the glycan-binding affinity and specificity of galectin-1. Regarding immunomodulation, CXCL4 significantly increases the apoptotic activity of galectin-1 on activated CD8[+] T cells, while no effect is observed in CD4[+] T cells. The opposite is found for another galectin-chemokine pair, i.e., galectin-9/CCL5. This heterodimer significantly reduces the galectin-9 induced apoptosis of CD4[+] T cells and not of CD8[+] T cells. Collectively, the current study describes an immunomodulatory mechanism in which specific galectin-chemokine interactions control the glycan-binding activity and immunoregulatory function of galectins.

[1] Amsterdam UMC location VUmc, Department of Medical Oncology, Cancer Center Amsterdam, Amsterdam, The Netherlands. [2] Laboratoire d'Ingénierie des Systèmes Macromoléculaires (LISM), Institut de Microbiologie de la Méditerranée (IMM), CNRS - Aix-Marseille Université, UMR7255 Marseille, France. [3] Amsterdam UMC location VUmc, Department of Radiation Oncology, Cancer Center Amsterdam, Amsterdam, The Netherlands. [4] Lund University, Department of Chemistry, Centre for Analysis and Synthesis, Lund, Sweden. [5] Lund University, Department of Laboratory Medicine, Section MIG, Lund, Sweden. [6] Maastricht University, Department of Biochemistry, Cardiovascular Research Institute Maastricht (CARIM), Maastricht, The Netherlands. ✉email: v.thijssen@amsterdamumc.nl

The human glycome is a complex and dynamic collection of cellular glycans and glycoconjugates that are involved in a plethora of cellular functions. Alterations in the glycan composition have been linked to different pathologies including genetic disorders, immune diseases and cancer[1]. In vertebrates, the information that is encoded by the glycome is deciphered by specific glycan binding proteins, called lectins. Interestingly, the complex repertoire of the glycome greatly exceeds the number of lectins[1]. This suggests that specific mechanisms exist to regulate and fine-tune lectin binding affinity in order to cover the broad spectrum of the glycome.

Recently, it was shown that galectin-3, a member of the galectin protein family, can form heterodimers with the chemokine CXCL12. While galectins are known to exert their function by binding to glycoconjugates[2], the interaction was found to be glycan-independent and to hamper CXCL12 activity[3]. Apart from galectin-3/CXCL12, the authors found interactions of other chemokines with either galectin-3 or galectin-1[3]. These observations support our previous finding that chemokine-like peptides can directly bind to different members of the galectin family[4,5]. We described that the heterodimerization induces changes in the galectin protein structure which affects the glycan-binding affinity and specificity of galectin-1[4,5]. More recently, modulation of glycan-binding activity was also observed in response to the interaction of galectin-1 with a specific fragment of the pre-B-cell receptor[6,7]. Collectively, these findings suggest that heterodimerization of galectins with other proteins could represent a mechanism to regulate galectin-glycan interactions. Such a mechanism would extend the functionality of this glycome-decoding protein family. However, whether chemokines could represent a protein family that regulates galectin function is not yet fully understood.

In the present study, we present evidence that the galectin-chemokine heterodimerization can indeed serve as a mechanism to modulate galectin glycan-binding and function. We found specific interactions between different chemokines and two immunomodulatory galectins i.e., galectin-1 and galectin-9. Focusing on the interaction between platelet factor 4 (PF4/CXCL4) and galectin-1, we show that the interaction modifies the glycan-binding properties of galectin-1 by inducing structural changes within its carbohydrate binding site. Moreover, we provide evidence that this interaction can modulate the immunomodulatory activity of galectin-1 as well as of galectin-9 on specific T cell subsets. Our findings provide evidence of galectin-chemokine cross-talk which can serve as a regulatory mechanism to guide and fine-tune glycoconjugate-galectin interactions leading to specific cellular responses.

## Results

**Several specific and glycan-independent galectin-cytokine heterodimers exist.** The current study aimed to explore whether galectin-chemokine interactions could serve as a mechanism to modulate galectin glycan-binding and function. First, we performed an interaction screen (Supplementary Fig. 1) and observed several specific interactions of galectin-1 and galectin-9 with different chemokines and cytokines. Based on our recent work showing complementary platelet responses induced by galectin-1 and platelet factor 4 (PF4 or CXCL4), we focused our investigation on this galectin-chemokine pair[8]. Spot blot analyses showed that CXCL4 binds galectin-1 to a similar extent as the positive control peptide, i.e., anginex[4] (Fig. 1a). To ensure that the interaction was not caused by changes in cytokine structure due to immobilization on the nitrocellulose filter, soluble galectin-1 with either CXCL4 or anginex were incubated in the presence of the crosslinking agent DSS. Subsequent gel-shift analyses

confirmed the expected shift in protein size of galectin-1 monomers (±14 kD) corresponding to the size of either CXCL4 (±8 kD) or anginex (±4 kD) (Fig. 1b). The intensity of the shifted band was weak suggesting interference of the bound chemokine with antibody binding to galectin-1. Of note, crosslinking with CXCL4 also revealed a band just above the galectin-1 dimer which could either represent a single galectin-1 monomer bound to two CXCL4 molecules or a galectin-1 dimer bound to a single CXCL4 molecule. To confirm that the interaction involved CXCL4 and was also not caused by structural abnormalities due to the crosslinking agent, excess heparin was added prior to crosslinking. Heparin, a known high affinity ligand of CXCL4, serves as a CXCL4 scavenger and indeed completely neutralized the galectin-1/CXCL4 interaction (Fig. 1c). Heparin did not cause a shift in galectin-1 size suggesting no interaction of both molecules. At the same time, the addition of excess lactose, a carbohydrate ligand of galectin-1, did not prevent the interaction (Fig. 1d). The latter indicated that CXCL4 does not directly compete for glycan binding in the carbohydrate recognition domain of galectin-1. Furthermore, the interaction was dose-dependent and extended beyond the galectin-1/CXCL4 pair (Fig. 1e and Supplementary Fig. 1).

To characterize the binding kinetics of the galectin-1/CXCL4 heterodimer formation, surface plasmon resonance analysis was performed. Similar as previously described for the galectin-1:anginex interaction[4], the galectin-1:CXCL4 interaction showed a 1:2 stoichiometry (single association followed by biphasic dissociation) with binding affinities in the high nanomolar and low micromolar range (Supplementary Fig. 2 and Table 1). This also suggests that the high molecular weight band observed in the gel-shift assay most likely represents a galectin-1 monomer bound to a CXCL4 dimer. Collectively, all these data provide clear evidence of a direct interaction between galectin-1 and CXCL4.

**Structural characterization reveals that CXCL4 binds outside the galectin-1 carbohydrate binding site.** To further characterize the galectin-1/CXCL4 interaction, chemical shift perturbation (CSP) mapping was performed using $^{15}$N-labeled galectin-1 with or without CXCL4 protein. $^1$H-$^{15}$N HSQC spectra (Fig. 2a) show that in the presence of CXCL4, several galectin-1 resonances experience CSPs (chemical shift deviations and peak intensity decrease), indicative of complex formation (Fig. 2b + Supplementary Fig. 3). Remarkably, perturbed resonances localize at two different sites on galectin-1 structure: (i) a surface at the edge of the galectin-1 ß-sandwich fold (opposite to the dimer interface) (Fig. 2c), and (ii) a surface at the carbohydrate binding site (CBS) (Fig. 2c). The first surface is defined by perturbed resonances belonging mainly to the ß6, ß8 and ß9 strands. Within the CBS, residues from strands ß3 to ß6 are perturbed upon CXCL4 interaction impacting not only the CBS core known to interact with the galactose unit but also the surrounding areas. However, our cross-linking data show that CXCL4 interaction does not compete with glycan binding activity of galectin-1, indicating that CXCL4 docks onto the perturbed surface of galectin-1 at the edge of its ß-sheets and that this binding induces changes inside the CBS. To further investigate the structural basis of this interaction we performed NMR data-driven docking to generate a structural model of the galectin-1/CXCL4 complex. Residues from galectin-1 experiencing CSPs upon CXCL4 interaction and localized at the edge of the ß-sandwich fold were defined as ambiguous restraints in the docking calculations. On the CXCL4 side, residues from the ß-sheet, homologous to the anginex peptide sequence, were also considered as ambiguous restraints. In the structural model showing the best HADDOCK score, the CXCL4 ß-sheet docks onto the galectin-1 surface

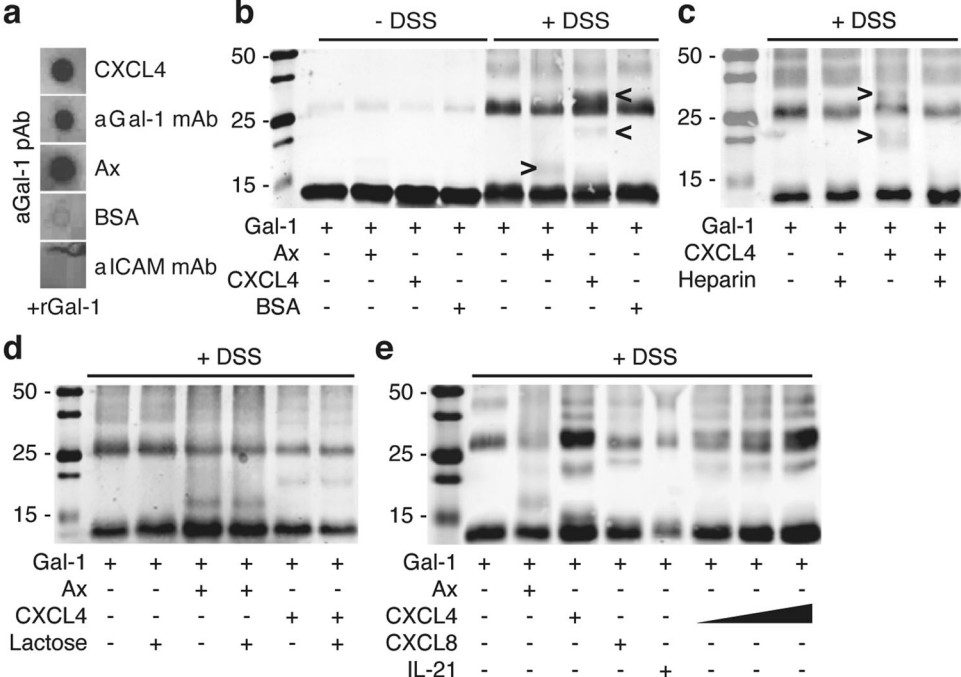

**Fig. 1 Specific protein-protein interactions occur between galectin-1 and different chemokines. a** Representative spot blot analysis. Indicated proteins were spotted onto nitrocellulose filter and following incubation with galectin-1 protein-protein interaction was revealed by staining with anti-galectin-1 antibody. Anginex (Ax) and anti-galectin-1 monoclonal antibody served as positive controls, bovine serum albumin (BSA) and anti-ICAM monoclonal antibody served as negative control. **b** Representative gel-shift analysis. Proteins (0.2 mg/mL) were incubated for 2 h in the presence or absence of cross-linking agent disuccinimidyl suberate (DSS). Standard Western blot with anti-galectin-1 polyclonal antibody. The arrowheads point towards a shift of galectin-1. **c** Similar as in **b** but in the presence or absence of excess heparin. **d** Similar as in **b** but in the presence or absence of excess β-lactose. **e** Similar as in **b** but in the presence of different cytokines or increasing concentrations of CXCL4.

**Table 1 Rate and affinity constants for galectin-1 binding.**

| Analyte | $k_a$ (Ms-1) | $k_{d1}$ (s-1) | $k_{d2}$ (s-1) | $K_1$ | $K_2$ |
|---|---|---|---|---|---|
| Anginex | $6.5 \times 10^3$ (±1.7) | $4.2 \times 10^{-2}$ (±0.4) | $5.9 \times 10^{-4}$ (±1.9) | $6.4 \pm 1.7\ \mu M$ | $90.0 \pm 6.7$ nM |
| CXCL4 | $1.02 \times 10^4$ (±0.05) | $9.2 \times 10^{-2}$ (±1.2) | $5.0 \times 10^{-3}$ (±0.5) | $9.0 \pm 0.8\ \mu M$ | $483 \pm 28$ nM |
| BSA | – | – | – | – | – |

The association rate constant ($k_a$) was obtained by plotting the $k_{obs}$ as a function of analyte concentration. $K_1$ is defined as $k_{d1}/k_a$ and $K_2$ is defined as $k_{d2}/k_a$. BSA bovine serum albumin. – No interaction detected.

involving mainly ß6 and ß9 strands (Fig. 2d). In addition, the CXCL4 C-terminal helix establishes contacts with strands ß8 and ß9 strands. To further validate the proposed structural model of the galectin-1/CXCL4 complex formation, the CXCL4 ß-sheet region (CXCL4[22-54]; Supplementary Fig. 4a, b) was produced after which the interaction with [15]N-labeled galectin-1 was tested. The perturbations observed on galectin-1 spectra corresponded to residues from the CBS and from the ß6 and ß9 strands (Supplementary Fig. 4c). When compared to CSPs induced by full-length CXCL4, the same regions of galectin-1 were affected with CXCL4[22-54] except for a few residues belonging to the ß8 and ß9 strands, which belong to the galectin-1 residues that interact with the CXCL4 C-terminal helix that is absent from CXCL4[22-54] (Supplementary Fig. 4d, e). Therefore, these data confirm the involvement of both the CXCL4 ß-sheet and its C-terminal helix in the interaction with galectin-1.

**Binding of CXCL4 to galectin-1 affects the glycan-binding affinity and specificity of galectin-1.** Interestingly, while interacting at the edge of the galectin-1 ß-sheet, CXCL4 induced

chemical shift perturbations within the galectin-1 CBS. Previously, we have shown that the direct protein-protein interactions can modulate the glycan binding affinity of galectin-1[5,7]. To explore whether binding of CXCL4 could also affect the carbohydrate binding affinity of galectin-1 we performed fluorescent anisotropy using a fluorescein-tagged high affinity tdga-probe[9]. Upon addition of 8 μM CXCL4, the baseline signal slightly increased, indicating only minimal interaction between CXCL4 and the probe. Subsequent titration of increasing concentrations of galectin-1 showed that the binding affinity of galectin-1 for this probe increased almost 10-fold to a $K_d$ of $0.09 \pm 0.01$ as compared to galectin-1 alone ($K_d$ of $0.88 \pm 0.05\ \mu M$) (Fig. 3a). In line with previous results[5], the $A_{max}$ (anisotropy of the galectin-probe complex) was also increased in the presence of CXCL4, which indicates a reduced mobility of the fluorescein tag. Similar as in the gel-shift experiments, the increased binding was neutralized by the addition of heparin (Fig. 3a). To further explore the modulation of glycan binding by CXCL4, comparable experiments were performed with a low affinity probe (lacto-N-tetraose; LNT-probe). While in the absence of CXCL4 no binding was observed due to very low $A_{max}$, the addition of CXCL4 induced a

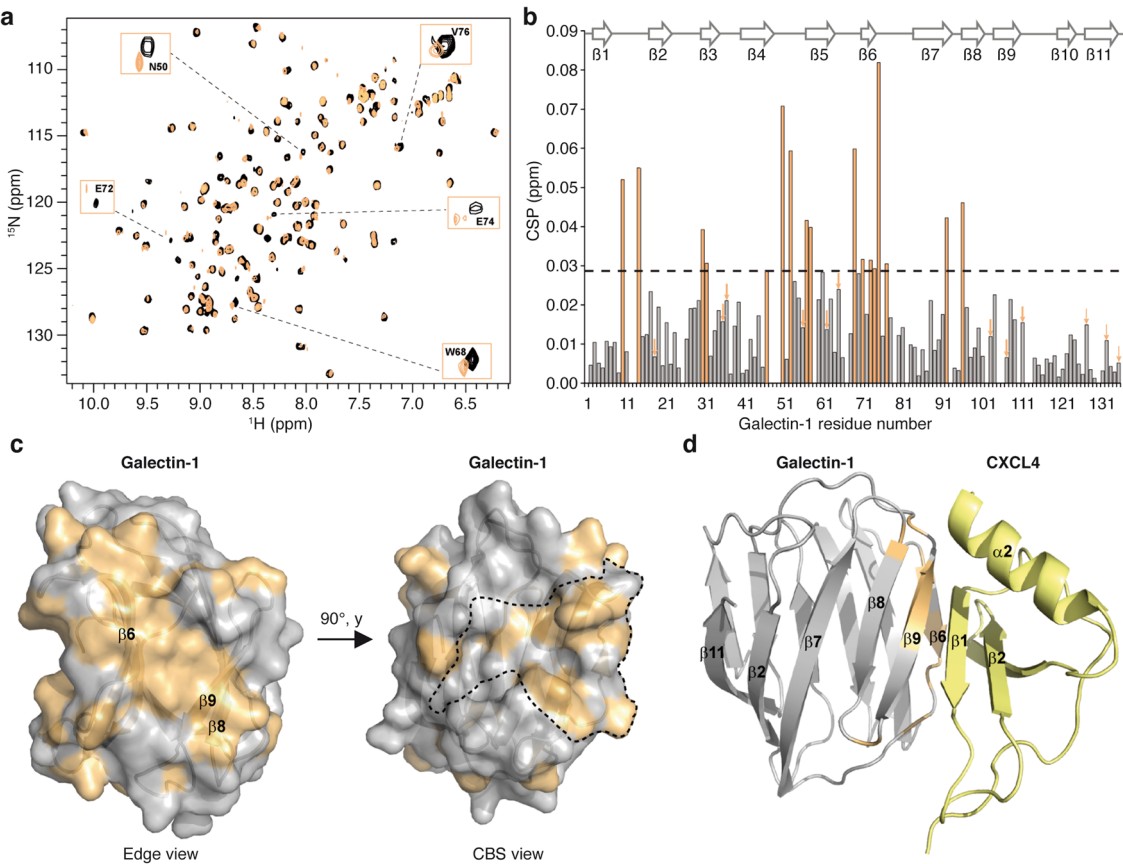

**Fig. 2 Structural characterization of the galectin-1/CXCL4 complex. a** Overlay of $^1$H-$^{15}$N HSQC spectra of $^{15}$N-labeled galectin-1 free (black trace) and bound to CXCL4 (light orange trace). **b** Histogram plot of CSPs observed for each galectin-1 resonance upon interaction to CXCL4. The dashed line represents 1σ from the average CSP, thereby defining the threshold selection for the most affected residues. Light orange bars correspond to CSP above the threshold. Arrows are pointing toward residue showing intensity decrease upon interaction to CXCL4 (Supplementary Fig. 3). Galectin-1 secondary structures are depicted above the plot. **c** Chemical shift perturbation mapping onto galectin-1 monomer structure which is shown as a semitransparent solvent-accessible surface with a ribbon model displayed below the surface. Residues presenting CSPs above the threshold (defined in **b**) are colored in light orange. Main ß-strands affected are labeled. Two galectin-1 orientations are presented: (i) the galectin-1 edge view, opposite to the dimer interface, and (ii) the CBS view with the CBS boundaries plotted in dotted lines. **d** The structural model of the galectin-1/CXCL4 complex. Galectin-1 is shown as grey ribbon whereas CXCL4 is colored yellow. Galectin-1 residues found at the interface of the complex and presenting CSPs above the threshold (defined in **b**) are colored in light orange.

measurable anisotropy and binding of the LNT-probe to galectin-1 with a $K_d$ in the range of the high-affinity tdga-probe, i.e., $0.48 \pm 0.18\,\mu M$ (Fig. 3b). At high galectin concentrations (>1.25 μM) the anisotropy levels decreased again, due to competition of the free galectin for the probe. The altered carbohydrate binding affinity was further confirmed by a competition experiment in which increasing concentrations of an unlabeled inhibitor of the galectin-1-probe interaction, i.e., the glycoprotein asialofetuin (ASF), was added to a fixed concentration of galectin-1 (the $EC_{50}$ of the corresponding slope in the absence of ASF). In the presence of CXCL4, the inhibitory potency of ASF was more than 600-fold increased (Fig. 3c). Of note, the effects were galectin-1/CXCL4 specific as CXCL4 did not affect glycan binding of e.g., galectin-3 nor did CCL5 and IL-8 affect glycan binding affinity of galectin-1 (Supplementary figure 5). Altogether, these results show that heterodimerization with CXCL4 can alter the affinity and specificity of galectin-1 for glycan binding.

**Galectin-chemokine interactions can modify the pro-apoptotic activity of galectins in immune cells.** To explore whether galectin-1/CXCL4 interactions represent an endogenous mechanism to control the function of galectin-1, we analyzed the

effect of the heterodimer formation on immune cells. We focused on T cell apoptosis since galectins are known to induce apoptosis in activated T cells[10–12]. Indeed, in line with previous findings, we did see time- and concentration-dependent induction of T cell (Jurkat) apoptosis by galectin-1 (Supplementary Fig. 6a). Interestingly, apoptosis induction by galectin-1 was significantly increased by CXCL4 and not by CCL5 or CXCL8 (Fig. 4a). The latter two were included since spot blot or gel shift assays indicated an interaction with galectin-1 as well (Fig. 1e and Supplementary Fig. 1). The effects of CXCL4 were dose-dependent (Fig. 4b) and could be blocked by the addition of either lactose, which competes for galectin-glycan binding, or heparin which scavenges CXCL4 (Fig. 4c). CCL5 (RANTES) showed an interaction with both galectin-1 and galectin-9 (Supplementary Fig. 1), but did not affect galectin-1 function. Therefore, we explored whether CCL5 could affect galectin-9-induced T cell apoptosis. Similar as for galectin-1, treatment with galectin-9 alone resulted in a time- and concentration-dependent induction of Jurkat apoptosis (Supplementary Fig. 6b). Interestingly, the induction of apoptosis by galectin-9 was specifically inhibited by CCL5 and not by CXCL4 or IL-8 (Fig. 4d). Again, the effect was dose-dependent (Fig. 4e) and could be neutralized by lactose (Fig. 4f).

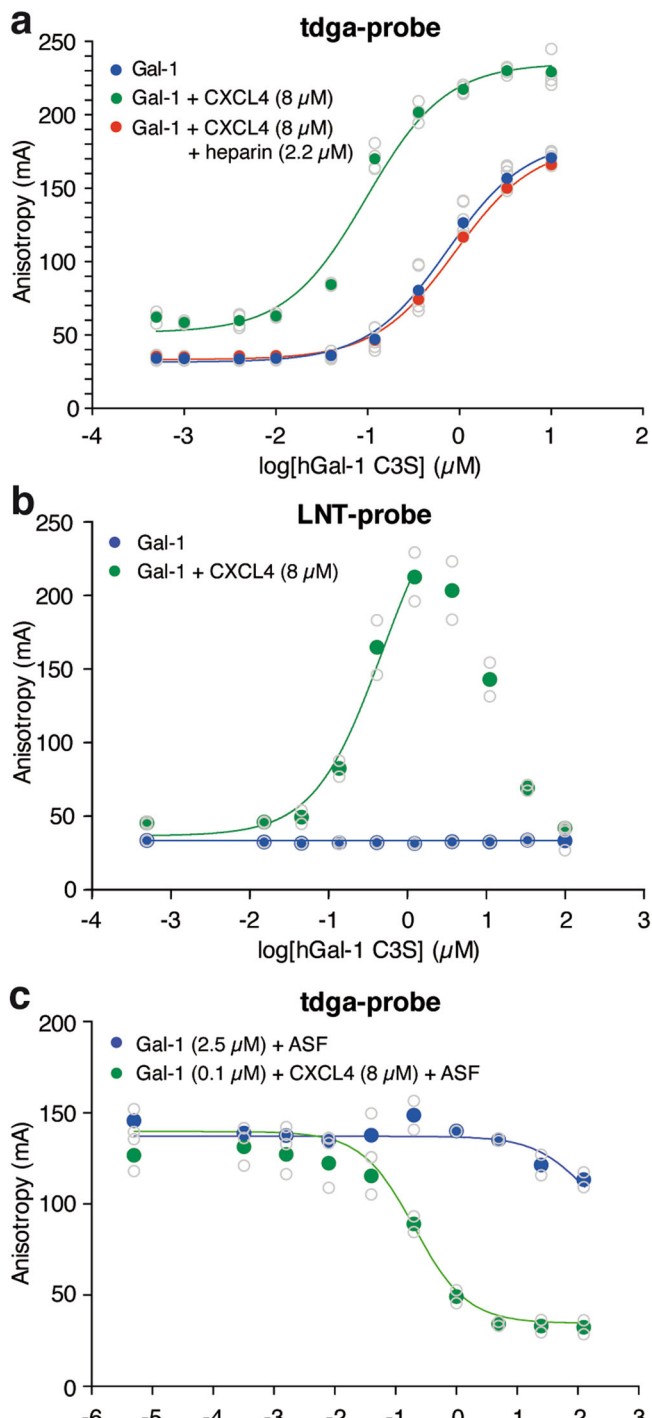

**Fig. 3 Interactions between galectin-1 and CXCL4 affect glycan-binding affinity and specificity of galectin-1. a** Anisotropy analyses using a fluorescently labeled high affinity thiodigalactoside amide (tdga)-probe with increasing galectin-1 concentrations in the absence (blue line, $N = 7$) or presence (green line, $N = 4$) of CXCL4 and excess heparin (red line, $N = 1$). **b** Anisotropy analyses using a fluorescently labeled low affinity lacto-N-triose (LNT)-probe with increasing galectin-1 concentrations in the absence (blue line, $N = 2$) or presence (green line, $N = 2$) of CXCL4. **c** Anisotropy analyses using a fluorescently labeled high affinity tdga-probe with fixed galectin-1 concentrations and increasing concentrations of an unlabeled high affinity asialofetuin (ASF)-probe in the presence (green line, $N = 2$) and absence (blue line, $N = 2$) of CXCL4.

These results suggest that heterodimerization with specific chemokines can have opposite effects on T-cell apoptosis induction by specific galectins.

**Regulation of apoptotic activity by galectin-chemokine heterodimers depends on immune cell type and activation status.** To further extend these findings, we next analyzed the effects of specific galectin-chemokine heterodimers on peripheral blood mononuclear cells (PBMCs) obtained from healthy donors. Firstly, we evaluated the effect of galectin-chemokine heterodimers on apoptosis in resting and activated PBMCs, the latter being more susceptible to galectin-induced apoptosis[12]. Cell activation was induced by PHA-L lectin treatment and confirmed by CD25 surface staining (Supplementary Fig. 7). In resting PBMCs, neither CXCL4 nor CCL5 affected the induction of apoptosis by galectin-1 and galectin-9, respectively (Fig. 5a). However, in activated PBMCs (1 μg PHA-L/mL), the pro-apoptotic effect of galectin-1 was significantly induced by CXCL4 while the pro-apoptotic effect of galectin-9 was significantly inhibited by CCL5 (Fig. 5b). These observations corroborated our findings in Jurkat cells. Of note, in the absence of galectins, neither CXCL4 nor CCL5 induced apoptosis in resting or activated PBMCs (Supplementary Fig. 8).

To determine whether the effects could be linked to specific immune cell subsets, analyses were performed on different immune cell subpopulations, i.e., CD3$^+$ T cells, CD3$^+$/CD8$^+$ cytotoxic T cells, CD3$^+$/CD4$^+$ T helper cells, and CD14$^+$ monocytic cells (For gating strategy see Supplementary Fig. 9). Regarding the galectin-1/CXCL4 heterodimer, we observed a significant increment in the percentage of apoptotic cells in CD3$^+$ cells (Fig. 5c) upon the addition of CXCL4 to galectin-1. Within the CD3$^+$ population, the increase of galectin-1-mediated apoptosis by CXCL4 could be mainly attributed to the CD8$^+$ cells while no effect was observed in CD4$^+$ cells (Fig. 5d). This suggests that within the analyzed T cell subsets, CD3$^+$/CD8$^+$ cells are particularly susceptible to galectin-1/CXCL4-induced apoptosis. The modulation of galectin-9 by CCL5 appears to display a different specificity regarding the modulation of immune cell survival. While CCL5 reduced the galectin-9-induced percentage of apoptotic CD3$^+$ cells (Fig. 5e), this was only significant for CD3$^+$/CD4$^+$ cells and not for CD3$^+$/CD8$^+$ cells (Fig. 5f). Comparable results were observed in CD14+ cells in which CXCL4 increased the effect of galectin-1 and CCL5 reduced the effect of galectin-9 (Supplementary Fig. 10). Altogether, these results suggest that galectin-chemokine heterodimers can alter the apoptotic activity of specific galectins on specific immune cell subpopulations.

## Discussion

In this study, we present evidence that specific galectins and chemokines can form heterodimers that affect the function of galectins through modifications of their glycan binding activity. In particular, we show that galectin-1/CXCL4 and galectin-9/CCL5 heterodimers can exert opposing immunoregulatory activity with regard to immune cell apoptosis. This reveals a cross-talk mechanism by which galectins and chemokines can jointly control the immune response.

Both galectins and chemokines are well established (immuno)regulatory protein families that exert a plethora of biological functions, including regulation of (immune) cell recruitment, growth, differentiation, and survival[13–15]. Thus far, both galectins and chemokines have been considered as distinct immunomodulatory protein families. However, there is increasing evidence that suggests a link between both. For example, it has been recently shown in two studies that galectin-3 can directly interact

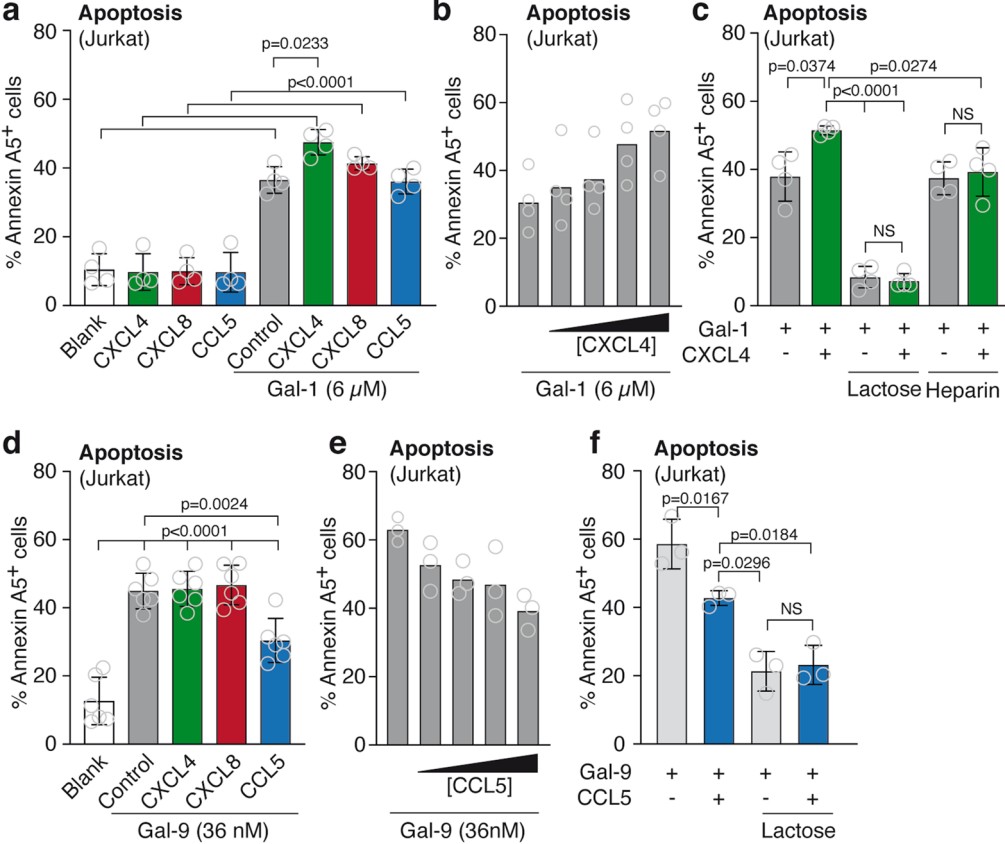

**Fig. 4 Specific galectin-chemokine interactions modify the pro-apoptotic function of galectins on Jurkat cells. a** Apoptosis as determined by FACS analysis (Annexin A5+ and PI+ staining). Jurkat cells were incubated for 1 h with 6 μM galectin-1 in the presence or absence of the indicated chemokines at equimolar concentrations. $N = 4$ independent experiments. **b** Similar as in **a** but with increasing concentrations of CXCL4 (0, 1.5, 3, 6 and 12 μM). $N = 4$ independent experiments. **c** Similar as in **a** but with the addition of either excess β-lactose or heparin. $N = 4$ independent experiments. **d** Apoptosis as determined by FACS analysis (Annexin A5+ and PI+ staining). Jurkat cells were incubated for 4 h with 36 nM galectin-9 in the presence or absence of the indicated chemokines at equimolar concentrations. $N = 6$ independent experiments. **e** Similar as in **d** but with increasing concentrations of CCL5 (0, 9, 18, 36, 72 nM). **f** Similar as in **d** but with the addition of excess lactose. $N = 3$ independent experiments.

with interferon-gamma[16] and CXCL12[3], thereby affecting their activity. While these findings show that galectin heterodimers can govern cytokine/chemokine function, our results show that the interaction can also regulate the function of galectin-1 and galectin-9. This adds an additional level to the immunomodulatory activity of these two galectins. As shown in the current study, a potential mechanism underlying the observed effects might involve modulation of the galectin glycan-binding affinity and specificity by inducing structural changes in the carbohydrate binding site. Previously, it has been shown that introducing mutations distal from the carbohydrate binding site can also result in small modifications in the structure of galectin-1 that affect glycan binding affinity[17]. In addition, we and others have previously reported on a close relationship between the galectin CRD structure and ligand binding[9,18–20]. In line with this, we have previously described that non-endogenous chemokine-like peptides can directly bind to galectins and affect glycan binding affinity[5]. More recently, we provided the first evidence that galectin-1 directly interacts with an endogenous protein, i.e., the pre-B cell receptor. This interaction also induces structural changes in galectin-1 which modulates glycan binding affinity, particularly with regard to LacNAc containing epitopes[6,7]. Our current findings extend these observations and suggest that the observed galectin-chemokine heterodimerization could represent a mechanism that allows the modification of the glycan-binding affinity and even specificity of galectin-1 and galectin-9. At the

same time, the exact mechanism underlying the glycan-binding effects of the galectin-cytokine interactions reported here is still unknown and might also involve, e.g., allosteric effects, as shown for other peptides/molecules[21–24]. Thus, while the exact reach and mechanisms of galectin-chemokine interactions remain to be explored, it is tempting to speculate that such interactions might be involved in adapting the glycan-binding repertoire of galectins. This is supported by our current results showing heterodimer formation of different chemokines with galectin-1 and/or galectin-9 as well as by a recent study by Eckardt et al. that identified several specific chemokine interactions with galectin-1 and/or galectin-3[3]. Of note, while some interactions were confirmed by both Eckhardt et al. and us, e.g., gal-1/CCL5, others were not, including gal-1/CXCL4. This is most likely related to differences in experimental setup and these findings warrant further studies to unravel the specific features that govern galectin-chemokine heterodimerization and to explore the functional consequences of these heterodimeric complexes.

Regarding the functional implications, we focused here on T cell apoptosis, instigated by the established role of galectin-1 and galectin-9 in this process[10–12]. For example, these two galectins, as well as others, can impair T cell function by inducing T cell apoptosis and skewing T cell differentiation towards a suppressive phenotype[10,11,25,26]. Our results confirm the previously described pro-apoptotic activity of galectin-1 and galectin-9 on T cells[10,11]. In addition, our current data show that this pro-apoptotic activity

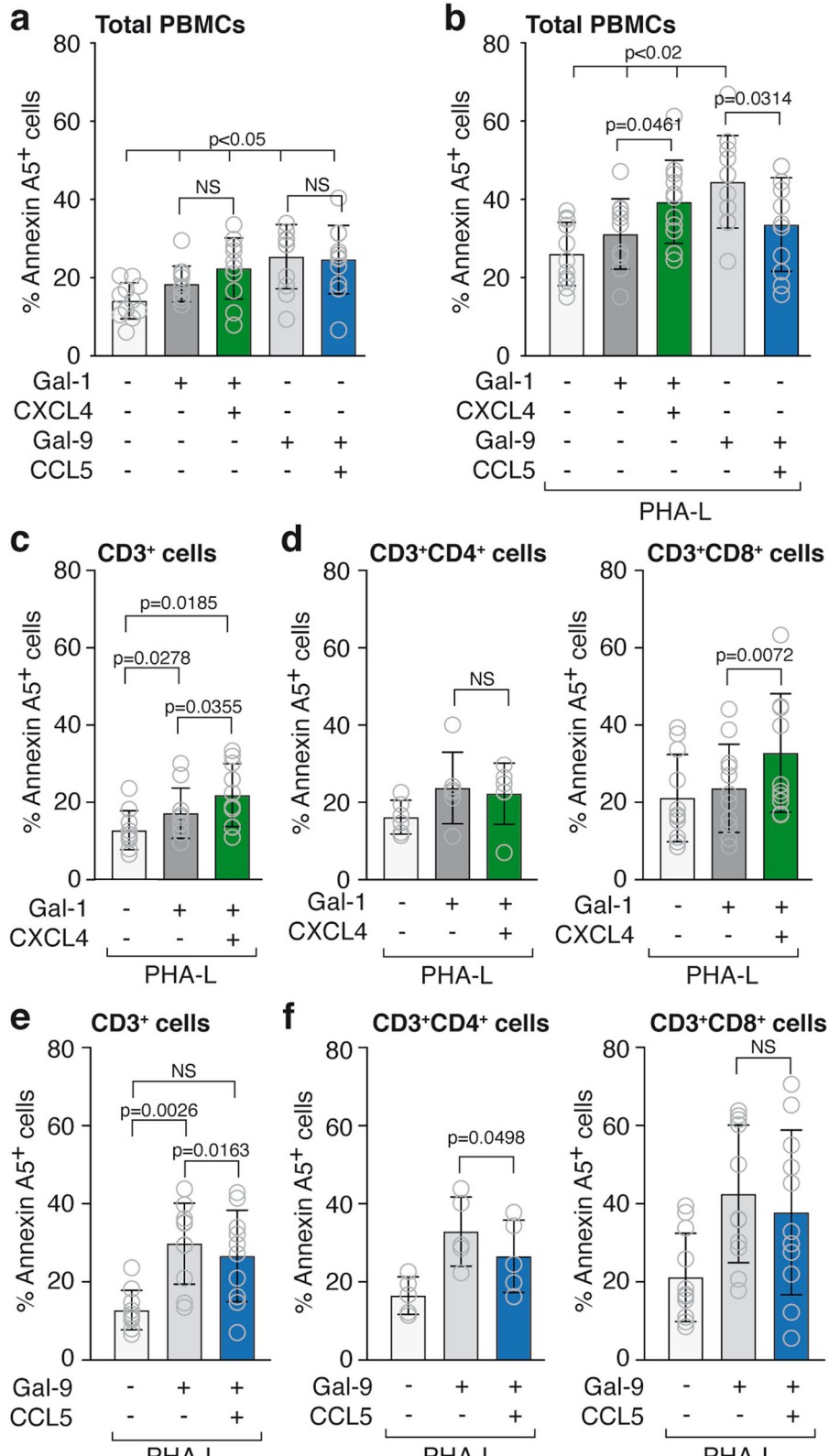

**Fig. 5 Specific galectin-chemokine interactions modify the pro-apoptotic function of galectins on specific immune cell subsets. a** Apoptosis as determined by FACS analysis (Annexin A5+). Peripheral blood mononuclear cells (PBMCs) were incubated for 24 h with 12 µM galectin-1 or 36 nM galectin-9 in the presence or absence of the indicated chemokines at equimolar concentrations. $N = 11$ different donors. **b** Similar as in **a** but PBMCs were activated for 24 h with 1 µg/mL phytohemagglutinin-L (PHA-L) prior to treatment with galectins and chemokines. $N = 11$ different donors. **c–f** Treatment similar as in **b** followed by FACS analysis with gating for specific subpopulations, i.e., CD3+, CD3+/CD4+, CD3+/CD8+. Graphs show percentage of Annexin A5+ cells within the indicated populations. $N \geq 6$.

can be either increased or inhibited by specific galectin-chemokine heterodimers, i.e., gal-1/CXCL4 or gal-9/CCL5, respectively. The mechanism(s) underlying this regulation should be further explored but it most likely involves an altered glycan-binding affinity of the specific galectin-chemokine heterodimers. For example, several T cell glycoproteins, e.g., CD3, CD7, CD43, CD45, have been linked to galectin-1-mediated T cell apoptosis[12,27,28]. In addition, it has been shown that the presence of specific glycan repertoires controls the ability of galectin-1 to induce apoptosis in specific T helper cells[10]. Thus, while speculative, it can be hypothesized that (changes in) the immune cell glycan signature together with altered glycan-binding affinity of galectin-chemokine heterodimers will determine whether or not certain immune cell subset will undergo apoptosis. The observation that the effects were more potent in Jurkat cells as compared to primary PBMCs could reflect the diverse glycosylation of different immune cell subsets in the latter. In line with this, the regulatory effects in PBMCs were only observed after PHA activation, suggesting a role for altered glycosylation patterns in activated vs. non-activated cells. Thus, future studies should thus aim at deciphering which immune cell subsets are particularly sensitive to specific galectin-chemokine heterodimers and whether this is linked to the cell surface glycan signatures and activation status.

Apart from further exploring the role of galectin-chemokine interactions in the regulation of immune cell apoptosis, our findings raise additional questions that should be addressed. For example, it is currently unknown to what extent specific galectin-chemokine heterodimerization affects other processes of the immune response, e.g., cell growth or cell differentiation/maturation. It has been shown that both galectin-1 and CXCL4 are involved in Th17 differentiation[29–31]. Whether the galectin-1/CXCL4 heterodimers stimulate or hamper this effect is unknown. In addition, next to immune cells, both galectin-1 and CXCL4 have been shown to regulate other cell types as well, e.g., endothelial cells during angiogenesis[4,32–34] and platelets during hemostasis[35,36]. Regarding the latter, we have recently shown that galectin-1 and CXCL4 colocalize on the platelet surface[8]. Since co-incubation did not affect binding of galectin-1 to platelets and the observed effects on platelet function appeared to be additive rather than synergistic, it appears that this specific heterodimer does not play a major effect in platelet homeostasis. Whether this is true for other cell types and other heterodimers remains to be studied. However, given our current observations, it can be assumed that the effects of certain galectin-chemokine heterodimerization extend beyond the immune cell compartment.

Our finding that galectin-chemokine interactions modulate glycan binding affinity and specificity could also have implications for the efficacy of glycan-based small molecule inhibitors that target galectins[37] and the growing arsenal of drugs that unleash the immune system against tumors[38,39]. It is essential to unravel the mechanisms by which the local tumor microenvironment (TME) controls the immune response and how these mechanisms interact[10,40]. Especially since tumor-controlled immunomodulation frequently involves impaired recruitment and/or function of immune cells due to the presence or absence of soluble factors like galectins and chemokines[41–43]. In that regard, the local balance between galectins, chemokines and their heterodimers could underlie the opposing effects seen in different experimental settings and biological contexts, e.g., the pro- or antiapoptotic activity of CXCL4[44–48]. This also links to the recent findings that galectins can serve as scavenger molecules for cytokines/chemokines, thereby hampering immune cell recruitment[3,16]. An additional intriguing but unexplored concept is whether galectins not only scavenge but also serve as chaperones for cytokines and chemokines by directing them, in a cytokine/chemokine-specific manner, to sites displaying a specific glycosylation pattern, e.g., glycoprotein cell-surface receptors. Further insight in the reciprocal relationship between galectins and cytokines/chemokines will therefore likely benefit the development of more effective treatment strategies.

A limitation of the current study is that we only explored the effects of specific galectin-chemokine combinations in vitro. To better understand the contribution of these interactions in different biological processes, in vivo experiments should be performed. However, to our opinion, this first requires further insight in the complex interactions between galectins and cytokines that might occur in vitro. For example, it has been suggested that heterodimers can also form between different members of the galectin family[49] or the chemokine family[50,51]. This suggests the existence of a complex regulatory network that links the local expression levels of galectins and chemokines to the complex glycome in order to regulate cellular functions in vivo. The width of such a network is still unknown and while the present in vitro study adds to the identification of this complex network, a full understanding of the extent of specific galectin-chemokine interactions and how they contribute to in vivo biology requires further research.

In summary, we present evidence that galectins and chemokines can directly interact and that this heterodimerization can affect the glycan-binding properties and immunoregulatory activity of galectins. These findings suggest the existence of a reciprocal mechanism to fine-tune the activity of galectins by chemokines. A better understanding of this mechanism will provide insight in the complex process of immunomodulation and will help the development of future (immuno)therapeutic strategies.

## Methods

Recombinant galectin-1 and galectin-9 were in-house produced or purchased from R&D systems. For fluorescent anisotropy and apoptosis assays, the oxidation resistant mutant galectin-1[52], i.e., galectin-1 C3S (in-house produced) was used.

The synthetic high-affinity tdga-probe (0.1 μM, 3′-[4-(fluorescein-5-amido-methyl)−1H-1,2,3-triazol-1-yl]−3′-deoxy-β-D-galactopyranosyl 3-(3,5-dimethoxybenzamido)−3-deoxy-1-thio-β-D-galactopyranoside)[9] as well as the low affinity probe (lacto-N-tetraose; LNT-probe)[53] were in-house produced.

Chemokines CXCL4 and CCL5 were either in-house produced[51] or purchased from R&D systems. All other chemokines and cytokines were purchased from R&D systems. The beta-sheet region, CXCL4[22-54], was chemically synthesized and purchased from Schafer-N. Anginex was a kind gift from Prof. K. Mayo (University of Minnesota, Minneapolis, USA). Antibodies were purchased from BD Biosciences unless indicated otherwise.

**Primary cells and cell lines**. The human leukemia cell line Jurkat (originally purchased from and STR authenticated by ATCC) was maintained in RPMI 1640, 2 mM glutamine medium (Gibco) supplemented with 10% heat-inactivated fetal bovine serum (Invitrogen), 100 U/mL penicillin and 100 μg/mL streptomycin (Invitrogen). Cells were cultured in a humidified incubator at 37 °C and 5% $CO_2$.

Peripheral blood mononuclear cells (PBMCs) were isolated from buffy coats from healthy donors (Purchased from Sanquin, no approval from ethical committee required) by density gradient centrifugation using Lymphoprep (Nycomed Pharma). Recovered cells were washed twice in sterile PBS with 0.1% bovine serum albumin and resuspended in RPMI-1640 2 mM glutamine (Gibco), containing 10% heat-inactivated fetal bovine serum (Invitrogen), 100 U/mL penicillin and 100 μg/mL streptomycin (Invitrogen). When indicated, PBMCs were treated with 1 μg/mL phytohemagglutinin-L (PHA-L, Invitrogen) for 24 h prior to stimulation with galectins/cytokines. All studies involving human samples were conducted following the Declaration of Helsinki principles and current legislation on the confidentiality of personal data.

**Spot blot**. Spot blot analysis was performed with 10 μL drops containing 3 μg of the specific proteins that were spotted onto a nitrocellulose membrane. The membrane was blocked with Odyssey blocking buffer (LI-COR Biosciences) for 1 h, washed with PBS/0.1% Tween, and incubated with 1 μg/mL recombinant galectin-1 or galectin-9 in PBS/0.1% Tween for 1 h. As control, the incubation step with recombinant galectin-1 was omitted. After washing with PBS/0.1% Tween, the membrane was incubated with rabbit anti-galectin-1 antibody (1:1000, Peprotech) or goat anti-galectin-9 antibody (1:250, R&D Systems) for 1 h. Following three

washing steps, the appropriate IRDye-labeled secondary antibody (1:10000, LI-COR Biosciences) was applied for 1 h. Finally, the membrane was washed twice with PBS/0.1% Tween and rinsed in PBS. Images were obtained by scanning the membrane with the Odyssey infrared imaging system (LI-COR Biosciences).

**Gel-shift analysis.** For gel-shift analysis, 2 µg of the specific proteins were combined in a total volume of 10 µL in PBS and incubated together with 2 µL DSS (Disuccinimidyl suberate, Thermo Scientific) for 2 h on ice to fix existing protein interactions. Where indicated, 25 mg/mL heparin or 5 mM lactose (Sigma Aldrich) was added. The reaction was quenched with 1 µL Tris buffer pH 7.5 for 15 m at room temperature. Subsequently, the protein mixture was suspended in Laemmli sample buffer (Biorad) supplemented with 1:20 β-mercapto-ethanol, boiled for 5 m and subjected to SDS-PAGE followed by Western immunoblotting using anti-galectin-1 antibody (1:1000, Peprotech). Following three washing steps, the appropriate IRDye-labeled secondary antibody (1:10000, LI-COR Biosciences) was applied for 1 h. Finally, the membrane was washed twice with PBS/0.1% Tween and rinsed in PBS. Images were obtained by scanning the membrane with the Odyssey infrared imaging system (LI-COR Biosciences).

**Surface plasmon resonance.** Binding kinetics between galectin-1 and CXCL4 were studied using the BIAcore 1000 biosensor system (BiaCore) as described previously[4]. In short, galectin-1 was covalently attached to CM5 Sensor Chips (BiaCore) using amine-coupling chemistry according to the manufacturer's instructions. Next, binding measurements were performed at 25 °C using HBS-EP buffer containing 10 mM HEPES, 150 mM NaCl, 3 mM EDTA and 0.005% surfactant P20 adjusted to pH 7.4. Interactions between galectin-1 and CXCL4 were analyzed by injection of different CXCL4 concentrations (20 µL at a flow rate of 30 µL/m). Flow cells were regenerated by injection of 20 µL regeneration buffer (10 mM glycine-HCl, pH 2.0). Association-rate ($K_a$) and dissociation-rate ($K_d$) constants were obtained by analysis of the sensorgrams using the Biaevaluation software, version 3.2. All measurements were performed at least in duplicate at all concentrations and the experiment was performed in duplo.

**Fluorescence anisotropy.** Fluorescence anisotropy analysis was performed as described previously[5,54]. For these experiments, the oxidation-resistant mutant galectin-1, i.e., galectin-1 C3S, was used to avoid the necessity to add β-mercaptoethanol, which may disturb interacting glycoprotein ligands[52,55]. In brief, 40 µL reaction mixture containing different concentrations of galectin-1 C3S and 0.1 µM of fluorescein labeled saccharide probes either with or without CXCL4 at the indicated concentrations was incubated under slow rotary shaking for 5 m in black 96-well half-area polystyrene microplates. All dilutions were prepared in HEPES buffer with pH 7.4 and in the absence of salt to prevent precipitation of CXCL4. Fluorescence anisotropy of fluorescein-tagged saccharide probes was measured with excitation at 485 nm and emission at 520 nm using a using a PheraStarFS plate reader with software PHERAstar Mars version 2.10 R3 (BMG). Binding curves were generated by plotting anisotropy against increasing concentrations of galectin-1. Anisotropy was measured without galectin-1 ($A_0$ representing free probe), at experimental conditions (A) and at saturating concentrations of galectin-1 ($A_{max}$). The experiments were performed at least in duplicate. For the inhibition assay, increasing concentrations of the glycoprotein asialofetuin (ASF), an inhibitor of the galectin-1-probe interaction, were added to fixed low concentrations of galectin-1C3S (0.1 µM) and tdga-probe in the presence or absence of 8 µM CXCL4. Data plotting, non-linear regression analysis and curve construction was performed with Prism 9 software (Graphpad).

**Nuclear magnetic resonance.** For NMR experiments, galectin-1 has been produced as [15]N-labeled protein as previously described[6] and prepared in 20 mM KPO$_4$ pH 5.6 buffer with 8% D$_2$O. All NMR spectra were acquired at 30 °C on a Bruker AVANCEIII spectrometer operating at 600 MHz and equipped with cold probe. NMR data were processed with TopSpin and analyzed with Ccpnmr Analysis[56]. For interaction experiments with CXCL4, [1]H-[15]N-HSQC spectra have been recorded on [15]N-labeled galectin-1 (40 µM) in the absence and in the presence 80 µM of CXCL4. The same experiments have been performed with [15]N-labeled galectin-1 (350 µM) in the presence of CXCL4[22-54] (700 µM). The following formula was used for extracting chemical shift perturbations (CSPs), which takes into consideration the [1]H and [15]N chemical shift range of NMR signals: CSP (ppm)$= =[\Delta\delta_{HN}{}^2 + (\Delta\delta_N{}^2/25)]^{1/2}$ where $\Delta\delta_{HN}$ and $\Delta\delta_N$ are, respectively, the proton and nitrogen chemical shift variations of each residue.

**NMR-data driven docking.** Models of galectin-1 in complex with CXCL4 were built with the version HADDOCK2.4 of "The HADDOCK web server for data-driven biomolecular docking"[57]. As starting structures, we used the structures of galectin-1 (PDB ID 2KM2) and CXCL4 (PDB ID 1F9Q). The galectin-1 and CXCL4 structures were then docked within HADDOCK. Residues of galectin-1 that showed CSPs above the threshold were considered as active Ambiguous Restraints (AIR). On the CXCL4 side, the beta-sheet residues homologous to the anginex[58] sequence were considered as AIR. Calculations were performed with 2000 structures during the HADDOCK rigid body energy minimization,

400 structures during the refinement, and 200 structures during the refinement in explicit water.

**Apoptosis assays.** For analysis of apoptosis, either $1 \times 10^5$ Jurkat cells or PBMCs per well were plated in 24-well plates and incubated for the indicated time and concentration with galectin-1 or galectin-9 alone or in combination with the indicated concentration of chemokines.

Following treatment, Jurkat cells were washed twice in ice-cold PBS and stained with (FITC)-conjugated annexin A5 (Immunotools) and propidium iodide, following the manufacturer's instructions. Cells were subsequently analyzed using a FACSCalibur instrument (BD Biosciences). When indicated, 0.1 M lactose (Sigma-Aldrich) or 0.1 M heparin (Leo Pharma) were added to the cell culture just prior to galectin/chemokine treatment. Experiments were performed at least in triplicate.

Following PBMC treatment, PBMCs were blocked during 20 m with 1% BSA/PBS at 4 °C, washed in ice-cold PBS and incubated during 30 m at 4 °C with a combination of fluorescently conjugated monoclonal antibodies against CD3, CD8, CD4 and CD14: Anti CD3-APC (BD Clone SK7), anti-CD8-V500 (BD Clone SK1), anti-CD4-PerCP Cy5.5 (BD Clone SK73), anti-CD14-PE (BD Clone MφP9). Subsequently, cells were washed twice with 0.1% BSA/PBS and stained with (FITC)-conjugated annexin A5 (Immunotools) following the manufacturer's instructions. Data acquisition was performed using a FACSCanto II instrument (BD Biosciences). Experiments were performed using PBMCs from at least six different donors. All data were analyzed using Kaluza analysis software (Beckman Coulter).

**Statistics and reproducibility.** Of all experiments, the individual data are presented together with the mean of at least three independent experiments unless indicated otherwise. To calculate statistically significant differences of the Jurkat and PBMC apoptosis data, the paired student's $t$-test and the one-way ANOVA with Tukey's multiple comparison tests were used. Statistical computations were performed using Graphpad Prism software (v8 or v9) and $p$-values < 0.05 were considered as statistically significant.

**Reporting summary.** Further information on research design is available in the Nature Research Reporting Summary linked to this article.

## Data availability

The data that support the findings of this study are available in the supplementary material of this article (Supplementary data). This includes the original Western blot scans from Fig. 1b–e and the data presented in Figs. 3–5.

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

## Acknowledgements

The authors thank K.H.M. (University of Minnesota) for providing anginex and Anita Stam and Barbro Kahl-Knutson for excellent technical assistance. Part of this research was supported by a grant of the Dutch Cancer Society (KWF11040 to VLT).

## Author contributions

L.S., I.S., P.T., R.H., E.A., K.C., V.T. conducted experiments and generated data. L.S., I.S., P.T., R.H., H.L., A.W.G., L.E., R.K., V.T. analyzed and interpreted data. TdG., U.N., H.L., A.W.G., L.E., R.K., V.T. conceived, designed and/or supervised experiments. L.S., I.S., H.L., L.E., V.T. wrote the main manuscript. All authors approved the final manuscript.

## Competing interests

The authors declare the following competing interests: H.L. and U.J.N. are shareholders in Galecto Biotech AB, a company developing galectin inhibitors. The remaining authors declare no competing interests.
