## [Transparent Peer Review File · Communications Biology]

Reviewers' comments:

Reviewer #1 (Remarks to the Author):

The authors claim to report "a novel immunomodulatory mechanism", chemokine-galectin interaction "potentiating" (would "increasing" do it, if it does?) apoptotic activity of two galectins, the case of galectin-9 not further studied on the structural level. Reading the first passage of the Introduction, it becomes clear that a previous report (ref. 3) had reported on chemokine(like peptide)-galectin recognition, followed by a report on modulation of glycan binding by peptide recognition in 2011. Cited work by the authors on Anginex (is it "chemokine-derived"?) certainly has impact on this study's level of novelty. Since these authors' publication in 2011, the special concept has not received support. Considering the evidence given in the literature, it is not correct to write about chemokines as regulators of galectin function to "remain unexplored", better "not yet fully understood", as the recent publication on CXCL12/galectin-3 shows. Also, the authors have previously published a report on platelets using CXCL4 and galectin-1 (PLoS ONE 2021;16(1):e0244736), with clear impact on novelty of this study, too.

The following points arise by study of the text: the authors should add information on the CXCL4 beta-sheet peptide 22-54 and the fluorescence-tagged probe production and source/application of galectin-9 antibody. Specificity controls for spot blots are missing (Fig. 1a, Fig. S1). Omission of galectin is an essential control for specificity. Instead of snap shots, whole membrane should be shown. On p. 10, SPR curves for 1:2 interaction should be shown, detailed crosslinking studies and product analysis done. Is there binding of heparin to Gal-1? Can the authors exclude CXCL4 binding to their probes? The selective inhibition of heterodi(tri)merization will essentially be required as proof for the assumption, as shifts in blots do not reach level to be convincing. The interpretation is complicated by 10% serum presence with all its constituents making assays rather complex. Note that CXCL4 is anti-apoptotic on many cell types (could be by binding growth factors and apoptotic mediators, e.g. JBC 1995;270:15059). It is in this context quite remarkable that now it behaves as pro-apoptotic regulator. What about the effect of CCL5 on galectin-9? Is the reduction in apoptosis by adding CCL5 in Fig. 4a significant (scale bar positions cast doubt about it)? Structural analysis on galectin-1 is required to clarify this. Opposite effects of chemokines on two galectins require further documentation. Data on primary cells have an anecdotal touch, mechanisms are not explored. Data about monocytes (in the text) should be shown in figures. The authors are advised to refrain from premature generalizations (implying relevance for every galectin; even the special case of galectin-1 and cooperativity is disputable, because the cited study on titration calorimetry is not in line with other studies that did not reveal the positive cooperativity reported in ref 25: here, the literature should be completely documented considering Hill plots etc.). Such evidence should also be provided to support the given claim at least with bilayer models.

Reviewer #2 (Remarks to the Author):

In this manuscript by Sanjurjo et al., the authors decipher the relationship between galectin family members with chemokines and their potential role in modulating immune regulation. The following are some of the concerns which needs to be addressed to give clarity and validation of the findings.

1. In figure 1, the authors use recombinant proteins to interaction with gel-shift assay between the chemokines and Galectin-1. Recombinant proteins seem to have much higher tendencies to interact in a cell free system, however this may not be the case within the cell. Could they show the same interactions with immunoprecipitation of Gal-1? In the gel shift assay, the bands they claim to be due to the interaction is really faint and difficult to ensure its specificity.

2. In Figure 4, authors show that CXCL4 addition increases apoptosis in Jurkat cells induced by Galectin-1. However, the dose of CXCL4 is in uM ranges, which is extremely high levels compared to what is produced in the cells. Hence, it is difficult to see the relevance of this results in the biological context especially at high doses of recombinant proteins.

3. The authors show that the CXCL4- Gal-1 interaction is not dependent upon the glycan binding activity, as it is not affected by addition of lactose, however, in the apoptosis induction assay, addition of lactose completely reverses apoptosis enhanced by CXCL4-Gal1. Could the authors explain why this could be?

4. In order to see if the effect on apoptosis interaction is specific and mediated through the chemokines, the authors need to show that the effect on apoptosis is abrogated in the presence of neutralizing antibodies for CCL5 or CXCL4.

5. Figure 5, in the case of PBMCs, the dose used was further increased to 12uM (Gal1), in spite of this the apoptosis induction was very moderate. It is also interesting that CD8+ T cells show no increase in apoptosis with Galectin-1 (c) or are not sensitive, however addition of CXCL4 increase apoptosis in these cells. However, the CXCL4 only control is missing in these experiments. Hence it is difficult to interpret if the increase is due to CXCL4 itself.

b) Few other studies have reported that galectin-1 mediated apoptosis was affected by the use of reducing agent in the culture media used to prevent the oxidation of galectin-1. Did the authors use recombinant-Gal1 with the reducing agent? Instead could they show these results using the oxidation resistant mutant Gal1?

6. Authors have not discussed the discrepancy in results in terms of Jurkat T cells and activated PBMCs, as the effects do not appear to be very consistent between the lines.

Reviewer #3 (Remarks to the Author):

The author presents an interesting view on how chemokines regulate the binding affinity and functionality of galectins. The author used NMR to characterize the binding site of CXCL4 and how it can modify galectin-1 structure which considered novel. Also It is interesting to show that galectins mediated versatile functions is not only controlled by the interacted ligand on cell surface but even controlled by the presence of specific chemokines in the microenvironment that bring extra layer of complexity. The article was convincing and easy to be read and understand

I have a minor comment to be addressed:

- In supplementary Fig1: the antibody used to detect galectin-1 shows cross interacted with galectin-9 which means that antibody is not specific for galectin-1 can the author had another blot showing specificity for galectin-1 over galectin-9.

We thank the reviewers for their time and contribution and the editorial team for providing us with the opportunity to improve the manuscript and submit a revised version.

We apologize for the time it took to prepare this revision. Please note that due to covid-19 restrictions we had only limited access to the lab and resources for quite some time.

Altogether, we are confident that we have been able to improve the manuscript based on the questions and suggestions of the reviewers. Additional experiments and data have been added and textual changes have been made to further support and clarify our findings. A detailed reply to the specific comments can be found below.

Detailed reply to specific comments

Reviewer #1 (Remarks to the Author):

The authors claim to report “a novel immunomodulatory mechanism”, chemokine-galectin interaction “potentiating” (would “increasing” do it, if it does?) apoptotic activity of two galectins, the case of galectin-9 not further studied on the structural level. Reading the first passage of the Introduction, it becomes clear that a previous report (ref. 3) had reported on chemokine-(like peptide)-galectin recognition, followed by a report on modulation of glycan binding by peptide recognition in 2011. Cited work by the authors on Anginex (is it “chemokine-derived”?) certainly has impact on this study’s level of novelty. Since these authors’ publication in 2011, the special concept has not received support. Considering the evidence given in the literature, it is not correct to write about chemokines as regulators of galectin function to “remain unexplored”, better “not yet fully understood”, as the recent publication on CXCL12/galectin-3 shows. Also, the authors have previously published a report on platelets using CXCL4 and galectin-1 (PLoS ONE 2021;16(1):e0244736), with clear impact on novelty of this study, too.

> While we value the reviewer's comment, we do not agree with the suggestion that previously published work affects the impact or novelty of the current study. Indeed, we and others have shown that synthetic peptides or peptide fragments can bind to galectins. However, we now provide evidence of an actual glycan-independent interaction between galectins and endogenous cytokines which is different from the paper the reviewer refers to. Moreover, we show for the first time that the interaction provides a mechanism by which chemokines can affect galectin glycan binding and galectin function. As referred to in our discussion, the papers on CXCL12/galectin-3 and gal-3/IFN γ show an opposite mechanism, i.e., regulation of cytokines by galectins (mainly by cytokine scavenging). Thus, we are the first to provide evidence of an unknown regulatory mechanism involving galectin-chemokine interactions and the functional consequences of this mechanism. Altogether, this provides sufficient impact and novelty to warrant publication in Communications Biology. To meet the reviewer in the middle of this argument, we have changed the wording 'potentiation' to 'increasing' and 'remain unexplored' to 'not yet fully understood' as suggested.

The following points arise by study of the text:

1. The authors should add information on the CXCL4 beta-sheet peptide 22-54 and the fluorescence-tagged probe production and source/application of galectin-9 antibody.

> The requested information on the probes, antibody and CXCL4 peptide has been added to the material and methods section. In addition, Supplementary Figure 3 (now **Supplementary figure 4**) has been modified to include the peptide amino acid sequence used as well as the location of the peptide sequence on the CXCL4 structure.

2. Specificity controls for spot blots are missing (fig. 1a, Fig. S1). Omission of galectin is an essential control for specificity. Instead of snap shots, whole membrane should be shown.

> The requested controls and whole membranes have been added to **supplementary figure 1**.

3. On p. 10, SPR curves for 1:2 interaction should be shown, detailed crosslinking studies and product analysis done. Is there binding of heparin to Gal-1?

> For SPR analysis, galectin-1 was crosslinked to the sensor chip as described before and as referred to in the M&M (Thijssen et al. PNAS 2006). For clarification, we have added the sensograms of the gal-1/CXCL4 SPR analysis to the supplementary data (**supplementary figure 2**).

The 1:2 interaction was revealed by analysis of the sensograms using the bivalence software. This showed the best fit with a 1:2 stoichiometry model. Moreover, since the gelshift data also suggested the presence of a single gal-1 monomer with two CXCL4 molecules attached (**figure 1c**) and CXCL4 is known to form multimers, we are confident these analyses are valid. Regarding the 1:2 stoichiometry obtained from SPR analyses, we have added the following sentence to the results sections linking this stoichiometry data to the gel-shift data:

'This also suggests that the high molecular weight band observed in the gel-shift assay most likely represents a galectin-1 monomer bound to a CXCL4 dimer.'

> Regarding heparin-binding, the results shown in **Figure 1b** as well as in **Figure 3a** show that heparin by itself does not bind to galectin-1 as this would have otherwise caused a noticeable gelshift or change in fluorescent polarization. We have added this observation to the result section as follows: Heparin did not cause a shift in galectin-1 size suggesting no interaction of both molecules.

4. Can the authors exclude CXCL4 binding to their probes?

> As shown in **Figure 3a** and **Figure 3b**, at the lowest concentration of galectin, there is a slight increase in anisotropy in the presence of 8 μM of CXCL4. However, if we correct for this (normalize to baseline) there is still a clear shift of the curve to the left with increasing gal-1 concentrations. To acknowledge this, we have rephrased the results section as follows:

Upon addition of 8 μM CXCL4, the baseline signal slightly increased, indicating only minimal interaction between CXCL4 and the probe. Subsequent titration of increasing concentrations of galectin-1 showed that the binding affinity of galectin-1 for this probe increased almost 10-fold to a K_d of 0.09 ± 0.01 as compared to galectin-1 alone (K_d of 0.88 ± 0.05 μM) (**Figure 3a**). As additional controls, we have performed similar experiments with gal-3 and CXCL4 as well as with gal-1 and CCL5 or IL-8 (**supplementary figure 5**). Here, a small effect on probe binding is also occasionally seen but no increase in probe affinity can be observed.

5. The selective inhibition of heterodi(tri)merization will essentially be required as proof for the assumption, as shifts in blots do not reach level to be convincing.

> While we understand that the reviewer considers the shifts in the spot blots not convincing as specific inhibition of heterodi(tri)merization, it is not the only evidence provided in this study. Additional evidence is provided by the fluorescent polarization experiments (**figure 3**) as well as by the apoptosis experiments (**figure 4**) with inhibition of the gal-1/CXCL4 interaction by heparin or of the gal-1/glycan interaction by lactose. Moreover, gal-1/CXCL4 complex formation is demonstrated by the NMR experiments where addition of CXCL4 to ^{15}N -labeled Gal-1 shows significant chemical shift perturbations (**figure 2 and supplementary figure 2**) for several Gal-1 resonances. These results not only validate the assumption of a gal1/CXCL4 complex but also give the structural basis of the recognition between the two proteins.

6. The interpretation is complicated by 10% serum presence with all its constituents making assays rather complex. Note that CXCL4 is anti-apoptotic on many cell types (could be by binding growth factors and apoptotic mediators, e.g., JBC 1995;270:15059). It is in this context quite remarkable that now it behaves as pro-apoptotic regulator.

> Indeed, CXCL4 can bind to other proteins, e.g., VEGFA, as described in the mentioned paper. At the same time, that paper also showed that CXCL4 can act independent of binding to growth factors (in this particular case VEGF121). In fact, the authors state that "PF4 can disrupt VEGF receptor mediated signal transduction using an unknown mechanism which does not interfere with VEGF121 binding." Since gal-1 is known to promote VEGFR signaling by receptor clustering (Croci et al. Cell 2014), the CXCL4/Gal-1 interaction might actually be part of this 'unknown mechanism'. Indeed, we did see increased inhibition of human umbilical vein endothelial cells when treated with both gal-1 and CXCL4 compared to galectin-1 alone (which is known to stimulate migration). No effect on

proliferation was observed (see figure below). However, since we did not focus on angiogenesis or VEGF signaling, we did not include these data in the current manuscript. If deemed relevant, we can add these findings to the manuscript.

More importantly, our data presented in **supplementary figure 4** clearly show that CXCL4 alone does not affect apoptosis of PBMCs and that the observed effects are therefore not related to CXCL4 binding to or affecting other factors in the medium/serum.

The observation that CXCL4 is pro-apoptotic rather than anti-apoptotic is in line with other findings in different cell types, including in immune cells, e.g., Deng et al. *Cancer Lett.* 2019, Gao et al. *Cancer Biol Ther.* 2014, Liang et al. *Haematologica.* 2013, Mayo et al. 2001. As such, our findings could underlie in part the diverse biological effects seen with CXCL4 as its activity most likely relies on the presence of proteins that it can affect, like galectin-1. We have added this insight to the discussion.

7. What about the effect of CCL5 on galectin-9? Is the reduction in apoptosis by adding CCL5 in Fig. 4a significant (scale bar positions cast doubt about it)? Structural analysis on galectin-1 is required to clarify this.

> It is not entirely clear to which reduction in apoptosis by adding CCL5 in **figure 4a** the reviewer refers to. **Figure 4a** shows that only CXCL4 increases Jurkat apoptosis by gal-1. In the presence of CCL5 the apoptosis levels are equal to the control (grey bar, which is gal-1 alone). In **figure 4d** we do show that CCL5 inhibits the apoptosis induced by galectin-9 (while CXCL4 has no effect). This inhibition is significant as indicated in the figure. These findings could be confirmed in specific T cell subsets as shown in **figure 5**.

8. Opposite effects of chemokines on two galectins require further documentation.

> Indeed, as our data show, the tested chemokines (CXCL4 and CCL5) display opposite effects regarding apoptosis induction. While opposite effects are known to occur (see also point 6), it is interesting to note that the opposite effects appear to work in the same 'direction', i.e., immunosuppression. As our data show, stimulation of apoptosis by gal-1/CXCL4 mainly affects cytotoxic (CD8+ cells) while inhibition of apoptosis by gal-9/CCL5 affects regulatory (CD4+ cells). Together, this would lead to a more immunosuppressed environment.

Obviously, much more research into the underlying mechanisms is required. As already mentioned in the discussion section, our findings raise many additional questions, e.g., whether the opposite effects are related to immune cell glycosylation patterns, whether glycosylation changes upon immune cell activation/differentiation can be linked to the response, whether the effects are influenced by chemokine heterodimerization (Nesmelova et al. *JBC* 2004, von Hundelshausen et al. *Blood* 2005) or what the effects are on other cellular functions, like, migration, differentiation, activation? All these questions need further documentation but go beyond the scope of the current paper as they represent a new field of extended research.

9. Data on primary cells have an anecdotal touch, mechanisms are not explored. Data about monocytes (in the text) should be shown in figures.

> We do not agree with the remark that the data on primary cells are anecdotal. These data are derived using PBMCs from multiple different donors, the experiments include the relevant controls, and the observed responses are therefore not just a 'case report' but responses that are consistently seen and confirmed in the Jurkat cell line. The underlying mechanism have not been explored as would go beyond the scope of this study (see also remark 8) and as already acknowledged in the discussion section. Future studies by us and others within the research community should address the implications of the current findings from different perspectives.

We apologize for not including the monocyte data, these have now been added to the supplementary figures (**supplementary figure 9**).

10. The authors are advised to refrain from premature generalizations (implying relevance for every galectin; even the special case of galectin-1 and cooperativity is disputable, because the cited study on titration calorimetry is not in line with other studies that did not reveal the positive cooperativity reported in ref 25: here, the literature should be completely documented considering Hill plots etc.). Such evidence should also be provided to support the given claim at least with bilayer models.

> We apologize for the confusion caused by the reference to Nesmelova et al. J Mol Bio 2010. It was not intended to use that reference as evidence to support positive cooperativity (the paper in fact suggest negative cooperativity induced by low affinity ligands) but was included to support the notion/hypothesis that structural changes in a galectin CRD can affect glycan binding. This is not only reported in reference 25 but also by us and others in more recent papers (Salomonsson et al., Biochem 2010; Chien et al., Molecules 2017; Romero et al., Glycobiology 2016). Nevertheless, we acknowledge that the exact mechanism underlying the glycan-binding effects of galectin-cytokine interactions reported here is still unknown and might also involve, e.g., allosteric effects, as shown for other peptides/molecules in different papers by the Mayo group (Dings et al., J Med Chem 2012; Dings et al., J Pharmacol Exp Ther 2013; Ermakova et al., Glycobiol 2013; Miller et al., ChemMedChem 2021). However, the effects observed and reported thus far are usually quite subtle while the magnitude reported in this paper is considerable (e.g Fig. 3C indicates an affinity enhancement of > 1000-fold for ASF in the presence of CXCL4).

To acknowledge these different insights/mechanisms and to indicate that the exact mechanism is yet to be revealed, we have rewritten part of the discussion including all this information and at the same time refraining from too many generalizations.

Reviewer #2 (Remarks to the Author):

1. In figure 1, the authors use recombinant proteins to interaction with gel-shift assay between the chemokines and Galectin-1. Recombinant proteins seem to have much higher tendencies to interact in a cell free system, however this may not be the case within the cell. Could they show the same interactions with immunoprecipitation of Gal-1?

> We agree that cell-free systems do not mimic the in vivo situation. That is why we started with showing the interaction in cell free systems and subsequently confirming the functional consequences in cellular systems in vitro. Of note, we do not think the interaction will occur within the cell but rather on the cell surface as we have previously shown in platelets that galectin-1 and CXCL4 can colocalize on platelets (Dickhout et al. PLoSONE 2021).

> Regarding the IP comment, immunoprecipitation has been tried previously (also with recombinant proteins alone) but this was unsuccessful due to the required antibodies. These are relatively large as compared to the heterodimers and binding of antibody interferes with heterodimerization similar as heterodimerization interferes with antibody recognition. The latter is also evident from our gel shift assays in which the interaction appears fairly weak, most likely because the galectin-CXCL4 complex is not well recognized by the antibody. We have acknowledged this now in the first paragraph of the results section.

2. In the gel shift assay, the bands they claim to be due to the interaction is really faint and difficult to ensure its specificity.

> See comment above. Crosslinking galectin-1 to CXCL4 affects antibody binding. That is why we proceeded from spot blots (which requires antibody) and gel-shift to SPR (antibody-free but immobilized gal-1) and finally to NMR to have antibody-free systems in which we could measure direct protein-protein interactions without any other interfering molecules/agents/carrier material. All these assays confirmed the interaction as did functional experiments like the anisotropy assays. We have performed additional anisotropy assays to further confirm the specificity and these data have been added to the manuscript (**supplementary figure 5**).

3. In Figure 4, authors show that CXCL4 addition increases apoptosis in Jurkat cells induced by Galectin-1. However, the dose of CXCL4 is in uM ranges, which is extremely high levels compared to what is produced in the cells. Hence, it is difficult to see the relevance of this results in the biological context especially at high doses of recombinant proteins.

> While the concentrations might appear as high, such concentrations can be achieved locally. In fact, functional dimerization of galectin-1 occurs with a K_d of ± 7 micromolar so such concentrations have to be reached in a biological context for galectin dimers to become active. Here, we used galectin-1 at 10 uM since this is commonly used by others for the same type of experiments (see also point 6).

4. The authors show that the CXCL4- Gal-1 interaction is not dependent upon the glycan binding activity, as it is not affected by addition of lactose, however, in the apoptosis induction assay,

addition of lactose completely reverses apoptosis enhanced by CXCL4-Gal1. Could the authors explain why this could be?

> Indeed, binding of CXCL4 to galectin-1 is not dependent on glycan binding as the interaction can not be prevented by lactose (**figure 1**). However, in the functional assays (apoptosis data), lactose is added to completely prevent the binding of galectin-1 to glycans on the immune cell surface and this therefore completely inhibits the ability of galectin-1 to induce apoptosis (irrespective of whether or not CXCL4 is bound). In fact, the observation that there is no residual apoptosis induction by CXCL4 alone in the presence of lactose further indicates that it is the galectin-CXCL4 interaction that is important and not the individual effects of galectin-1 and CXCL4.

5. In order to see if the effect on apoptosis interaction is specific and mediated through the chemokines, the authors need to show that the effect on apoptosis is abrogated in the presence of neutralizing antibodies for CCL5 or CXCL4.

> Indeed, showing specificity of the galectin-cytokine interaction with regard to apoptosis induction is relevant. That is why we included the experiments with heparin (figure 3 and 4) since heparin blocks the gal-1/CXCL4 interaction. As shown, heparin prevents the increase of apoptosis induced by CXCL4 while gal-1 is still active (Figure 4c). To add further proof of specificity we performed additional fluorescent polarization studies showing lack of activity of CXCL4 on galectin-3 or CCL5/IL8 on galectin-1 (see also question 2).

6. Figure 5, in the case of PBMCs, the dose used was further increased to 12uM (Gal1), in spite of this the apoptosis induction was very moderate. It is also interesting that CD8+ T cells show no increase in apoptosis with Galectin-1 (c) or are not sensitive, however addition of CXCL4 increase apoptosis in these cells. However, the CXCL4 only control is missing in these experiments. Hence it is difficult to interpret if the increase is due to CXCL4 itself.

> Indeed, for apoptosis assays using PBMCs we used 12 microM. This concentration, as well as the observed apoptotic effects, are perfectly in line with previous studies that reported on the apoptotic activity of galectin-1 on PBMCs (Perillo et al. 1995 Nature, Toscano et al. 2007 Nature Immunology, Bi et al. 2008 JBC). Therefore, we are confident that our observations are valid and in agreement with previous findings. As shown in **supplementary figure 8**, the addition of CXCL4 alone has no effect on apoptosis of activated PBMCs. Thus, the increased apoptosis by adding CXCL4 to galectin-1 is not caused by apoptotic activity of CXCL4 by itself but relies on the presence of galectin-1.

7. Few other studies have reported that galectin-1 mediated apoptosis was affected by the use of reducing agent in the culture media used to prevent the oxidation of galectin-1. Did the authors use recombinant-Gal1 with the reducing agent? Instead, could they show these results using the oxidation resistant mutant Gal1?

> As indicated in the M&M, all the fluorescent anisotropy and apoptosis experiments were performed using the oxidation resistant galectin-1 mutant. So, no reducing agents were added that could have affected the results. For clarity, we have added the reference to this oxidation resistant mutant galectin-1 to the M&M.

8. Authors have not discussed the discrepancy in results in terms of Jurkat T cells and activated PBMCs, as the effects do not appear to be very consistent between the lines.

> We are not sure to which discrepancy the reviewer refers. The only possible discrepancy is that the cytokines appear to have no regulatory effect on non-activated PBMCs while effects are observed in Jurkat cells. However, activation of PBMCs by PHA-L results in similar effects as observed in Jurkat, i.e., potentiation of galectin-1-mediated apoptosis by CXCL4 and inhibition of galectin-9-mediated apoptosis by CCL5. Since this suggests that activation status affects susceptibility, we have added a

remark to the discussion that future research should aim at deciphering the link between susceptibility to galectin-cytokine pairs and activation status.

Reviewer #3 (Remarks to the Author):

The author presents an interesting view on how chemokines regulate the binding affinity and functionality of galectins. The author used NMR to characterize the binding site of CXCL4 and how it can modify galectin-1 structure which considered novel. Also, it is interesting to show that galectins mediated versatile functions is not only controlled by the interacted ligand on cell surface but even controlled by the presence of specific chemokines in the microenvironment that bring extra layer of complexity. The article was convincing and easy to be read and understand.

I have a minor comment to be addressed:

1. In supplementary Fig1: the antibody used to detect galectin-1 shows cross interacted with galectin-9 which means that antibody is not specific for galectin-1 can the author had another blot showing specificity for galectin-1 over galectin-9.

> Indeed, the galectin-1 antibody appears to show weak cross-reaction with galectin-9. However, one should consider that an excess of antibody (3 micrograms) was spotted onto the membrane and two additional antibody incubations were performed afterwards, so the observed signal should be considered as background signal. Therefore, only signals that are clearly above background were considered to be of interest. Furthermore, the spotblots were used as a quick first screen and confirmation of interaction was performed in antibody-free systems (like SPR analysis, NMR and fluorescent anisotropy) to have multiple levels of evidence for interaction. We have added the following text to the legend of **supplementary figure 1** to acknowledge this:

Of note, occasionally some weak background signal was observed, e.g., mAb gal-1/gal-9, mAb ICAM-1/gal-1, IL17a/gal-9. This was probably due to excess protein blotting and multiple antibody incubation steps, and therefore, these low signals were not considered as true interactions but as background signal.

For additional comparison of the signal, we have now added the whole membranes (see also response to comment 2 of reviewer 1.

REVIEWERS' COMMENTS:

Reviewer #2 (Remarks to the Author):

Thank you for addressing the concerns/comments.

Reviewer #3 (Remarks to the Author):

I thank the author for replying to the comments.